# Controllable modulation of precursor reactivity using chemical additives for systematic synthesis of high-quality quantum dots

Joonhyuck Park [1], Arun Jayaraman [1], Alex W. Schrader [1], Gyu Weon Hwang [2] & Hee-Sun Han [1,3,4 ✉]

The optical and electronic performance of quantum dots (QDs) are affected by their size distribution and structural quality. Although the synthetic strategies for size control are well established and widely applicable to various QD systems, the structural characteristics of QDs, such as morphology and crystallinity, are tuned mostly by trial and error in a material-specific manner. Here, we show that reaction temperature and precursor reactivity, the two parameters governing the surface-reaction kinetics during growth, govern the structural quality of QDs. For conventional precursors, their reactivity is determined by their chemical structure. Therefore, a variation of precursor reactivity requires the synthesis of different precursor molecules. As a result, existing precursor selections often have significant gaps in reactivity or require synthesis of precursor libraries comprising a large number of variants. We designed a sulfur precursor employing a boron-sulfur bond, which enables controllable modulation of their reactivity using commercially available Lewis bases. This precursor chemistry allows systematic optimization of the reaction temperature and precursor reactivity using a single precursor and grows high-quality QDs from cores of various sizes and materials. This work provides critical insights into the nanoparticle growth process and precursor designs, enabling the systematic preparation of high-quality QD of any sizes and materials.

[1] Department of Chemistry, University of Illinois at Urbana-Champaign, 600 South Mathews Avenue, Urbana, Illinois 61801, USA. [2] Center for Neuromorphic Engineering, Korea Institute of Science and Technology, Seoul 02792, Korea. [3] The Carl R. Woese Institute for Genomic Biology (IGB), University of Illinois at Urbana-Champaign, 1206W Gregory Drive, Urbana, Illinois 61801, USA. [4] Center for Biophysics and Quantitative Biology, University of Illinois at Urbana-Champaign, 600 South Mathews Avenue, Urbana, Illinois 61801, USA. ✉email: hshan@illinois.edu

With their size-dependent electronic properties, solution processability, and synthetic tunability, quantum dots (QDs) have been heavily studied for both fundamental research and commercial applications. Fundamental studies have revealed how the structural parameters of nanocrystals, such as size[1,2], shape[3,4], shell thickness[5,6], and ligands[7–9], influence the electronic structure of QDs. The precise control of the electronic structure of QDs renders them attractive for a broad range of applications, including light-emitting devices[10,11], quantum information processing[12,13], and bioimaging[14]. A critical requirement for high-performance QD devices is a QD material with a well-defined electronic structure, which requires tight size distribution and high structural quality. A tight size distribution ensures that each QD has uniform electronic characteristics, resulting in a narrow emission profile[15] and homogeneous band structure throughout the QD layer[16]. Uniform size also allows for a controlled assembly of nanoparticles into a higher-order structure[17]. Size control of QDs has been extensively studied and a controlled growth based on the LaMer model yields QDs with less than a single monolayer (ML) difference in their size variation[18,19].

Other than the size, the structural quality of nanocrystals also affects the electronic structure of the QD ensemble. The structural quality of crystals is often assessed by crystallinity, morphology, and a controlled interface structure. Crystallinity is a direct measure of the density of lattice defects, which often serve as sub-bandgap trap states[20]. These poorly defined electronic defects promote the non-radiative decay process, negatively impacting the quantum yield (QY) of QDs[21–23]. Shape-wise, spherical QDs are often preferred for QD-based devices. Previous studies confirm that spherical QDs yield improved optical and electronic properties compared to faceted QDs both in solution[24] and solid-state devices[25]. Especially for core-shell QDs, the spherical shape ensures uniform shell passivation, resulting in well-confined carrier wave functions[26]. The core-shell interface is another factor affecting the electronic structure of QDs. For instance, a gradient shell creates a soft confinement potential, suppressing non-radiative Auger decay, thereby minimizing blinking[27,28]. Despite the importance of the structural quality, no universal method yet exists to grow highly crystalline, spherical QDs having a desired core-shell interface structure. Instead, the growth condition is mostly optimized by trial and error, and an optimal condition for one QD system cannot be adapted to QDs of different sizes or materials.

Systematic growth of high-quality QDs, both core-only and core-shell QDs, requires careful optimization of reaction temperature and precursor reactivity. Reaction temperature governs the surface instability of QDs and lattice reconfiguration. High instability of surfaces promotes particle dissolution and undesired alloying of the core-shell materials. To avoid excessively labile surface, the growth temperature must be lower than the melting temperature ($T_m$) of nanocrystals. On the other hand, high-temperature growth is preferred to promote thermal annealing of defects and create spherical QDs[5,25]. Precursor reactivity is another parameter to be tuned. It governs reaction kinetics at the surface, thereby impacting the crystallinity and morphology of the resulting crystals. Previous studies on thin film growth confirm that the reactivity of precursors greatly affects the quality of the surface; highly reactive precursors produce rough surfaces[29,30], crystalline defects[31], or irregular grains[32,33]. For nanocrystal growth, however, precursor reactivity has been mostly considered to achieve size focusing[34], not to improve the structural quality. These observations emphasize that high-quality QD growth requires both reaction temperature and precursor reactivity to be optimized. For existing precursors, their reactivity is determined by their chemical structure and variation of precursor reactivity requires synthesis of new precursor molecules. Naturally, in most cases, the existing precursor selections have significant reactivity gaps. For instance, metal precursors often lack precursors having intermediate reactivity (e.g., diethylzinc vs. zinc(carboxylate)$_2$). Recently, researchers have made progress toward a systematic modulation of precursor conversion rate by synthesizing a library of molecules containing different organic substituents[19,35–37]. However, this approach requires synthesis of a large number of precursor variants, which is labor intensive and time consuming; furthermore, we learned that the reactivity range covered by the recently developed sulfur precursor library is not optimal for QD growth (see Supplementary Results).

Here we introduce an organoboron-based sulfur precursor whose reactivity can be predictably modulated using

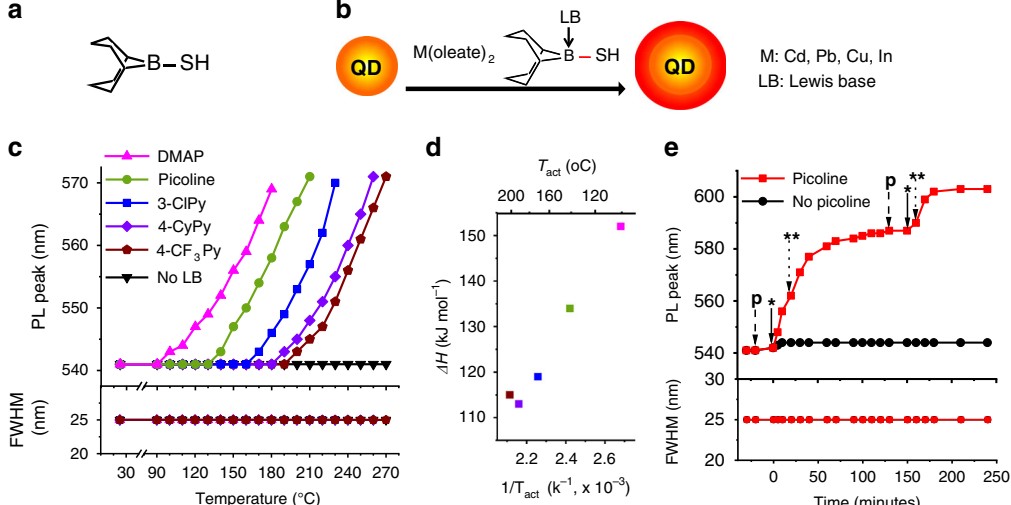

**Fig. 1 Characterization of [BBN-SH:LB]-based shell growth. a** Chemical structure of BBN-SH. **b** A schematic representation of shell deposition using BBN-SH. **c** Identification of the activation temperature ($T_{act}$) of BBN-SH upon the addition of LBs by monitoring CdS shell growth. $T_{act}$ of BBN-SH varies with the BF$_3$ affinity of LBs: DMAP (magenta triangle), picoline (olive circle), 3-ClPy (blue square), 4-CyPy (purple diamond), and 4-CF$_3$Py (brown pentagon). **d** LBs having higher BF$_3$ affinity ($\Delta H$) yield lower $T_{act}$. **e** LB is consumed during shell growth (p: precursor injection point, start (*) and stop (**) of picoline injection). QD emission remains tight during shell growth.

commercially available chemical additives. With its ability to fine-tune the precursor reactivity in a temperature-independent manner, the precursor can serve as a universal precursor for growing high-quality QDs of various materials and sizes.

## Results

**Design principle of the precursor chemistry**. We designed a sulfur precursor (R–B–SH) to enable the chemical-based modulation of precursor reactivity (Fig. 1a). A B–S bond is employed, because the bond length between boron and the adjacent atom can be altered upon the coordination of a Lewis base (LB) to boron[38]. We expect that LBs with a higher affinity to boron weaken the B–S bond to a greater extent, increasing the reactivity of the precursor. The Lewis affinity of organoborons has been characterized by the enthalpy change ($\Delta H$) upon complex formation between $BF_3$ and LBs[39]. Common LBs for $BF_3$ include amines, amidines, ketones, and ethers. For our studies, pyridine derivatives are used, because: (1) we can remotely vary the electron-donating strength through substituents on the pyridine ring, (2) they have a wide range of $BF_3$ affinities, and (3) they are commercially available. By varying organic substitutes on the pyridine ring with electron donating or electron withdrawing groups, we modulate their $BF_3$ affinities significantly (113–152 kJ $mol^{-1}$, Supplementary Table S1). The R group chosen for our precursor is 9-borabicyclo[3.3.1]nonane (BBN). This group is widely used in organoborane chemistry and their locked conformation ensures low steric hindrance around boron. A multi-gram quantity of the newly designed precursor, 9-mercapto-BBN (BBN-SH), is obtained with 89% yield via simple one-step synthesis (see "Methods"). We confirmed the coordination of a pyridine derivative, picoline, to the boron atom in BBN-SH by observation of an up-field peak shift in $^{11}B$ NMR (Supplementary Fig. S1).

**Chemical-based modulation of BBN-SH reactivity**. We confirmed that the reactivity of BBN-SH precursor can be predictably modulated by varying the $BF_3$ affinity of LBs. Precursor reactivity is characterized by monitoring the temperature at which the CdS shell growth is initiated (Fig. 1b). We referred to this as activation temperature ($T_{act}$). Briefly, a reaction vessel containing CdSe cores, precursors, and LB is slowly heated while monitoring the QD photoluminescence (PL) (Fig. 1c). The growth of CdS shells results in delocalization of the electron wave function into the shell layer, causing a redshift of the PL peak. LBs used for this study include dimethylaminopyridine (DMAP), picoline, 3-chloropyridine (3-ClPy), 4-(trimethylfluoro)-pyridine (4-$CF_3$Py), and 4-cyanopyridine (4-CyPy), in the decreasing order of their $BF_3$ affinity (Supplementary Table S1). As shown in Fig. 1c, the $T_{act}$ of BBN-SH can be tuned over a wide range (100–200 °C) by adding different LBs. As expected, pyridine derivatives with higher $BF_3$ affinity result in lower $T_{act}$, corresponding to increased reactivity of BBN-SH (Fig. 1d). However, between 4-CyPy and 4-$CF_3$Py, which has a marginal difference in $\Delta H$ (113 vs. 115 kJ $mol^{-1}$), 4-$CF_3$Py yields higher $T_{act}$. We attribute this reversed trend to steric hindrance by the BBN group. In all cases, the PL of QDs remains narrow (25 nm) during shell growth, indicating highly monodisperse size distribution. We then compared the rate of shell growth at a constant temperature in the presence of different LBs (Supplementary Fig. S2). As expected, LBs yielding higher reactivity of BBN-SH (or lower $T_{act}$) induce faster shell growth. We verified that non-pyridine-based LBs, such as 1,4-diazabicyclo [2.2.2]octane (DABCO), 1-methylpyrrolidine, and 1-phenylimidazole, modulate the reactivity of BBN-SH in the same way as the pyridine-based LBs (Supplementary Fig. S3). Not every LB can be used to modulate the reactivity of BBN-SH. Our growth results show that LBs having high steric hindrance around the coordinating atom (e.g., trioctylphosphine (TOP)) or hydrogen-containing amines (e.g., oleylamine (OAm)) cannot modulate the reactivity of BBN-SH. Of note, shell growth is not initiated even at high temperatures (>300 °C) in the absence of a LB. This observation reiterates the unique shell growth mechanism of our precursor chemistry: chemically induced precursor conversion. In contrast, existing shell growth chemistries involve heat-activated shell growth.

During growth, BBN-SH is quantitatively converted into shells. Shell thickness measured by transmission electron microscopy (TEM) images matches the expected MLs based on the injected amount of precursors. We then examined whether LB acts as catalysis or is consumed after precursor conversion. To test this, we first deposited a CdS shell on CdSe cores and waited for PL peak shift to the plateau, indicating the termination of shell growth. Then, fresh precursors were injected but the PL of QDs did not change. PL shift is reinitiated upon the injection of fresh picoline, confirming that LBs are consumed during the reaction (Fig. 1e).

**Limitations of conventional precursors and suboptimal quality of QDs**. High-quality shell deposition is achieved when (1) growth is carried out at a temperature that is slightly lower than the $T_m$ of the cores and (2) the reactivity of precursors is tuned so that precursors initiate QD growth at a temperature that is slightly lower than the growth temperature. Growth temperature should be set slightly below the $T_m$ of nanoparticles to prevent surface atoms from becoming excessively labile, while still promoting the effective elimination of structural defects via thermal annealing. Of note, the $T_m$ of nanoparticles varies significantly depending on their size and material (Supplementary Fig. S4). Accordingly, QDs of various sizes or materials require significantly different growth temperatures[40,41]. The requirement for precursor reactivity is imposed to induce controlled, defect-free growth without generating satellite particles. Optimizing precursor reactivity using the existing precursors, however, is difficult due to several reasons. First, there is no universal precursor whose reactivity can be tuned to achieve optimal conversion rates at a wide range of temperatures. Instead, different molecules must be synthesized to vary precursor reactivity. Second, no systematic method exists to identify a precursor that yields an optimal surface-reaction rate at a specific reaction temperature, leading to a trial-and-error-based selection of precursors. Third, the existing precursor libraries often have either significant gaps in reactivity or do not cover a wide enough range of reactivity to accommodate vastly different growth temperature required for QDs of different sizes and materials[40,41]. Due to these limitations, current synthetic methods fail to grow high-quality QDs of a wide range of sizes and materials, for both core-only and core-shell types.

The most commonly used sulfur precursors are octanethiol (OT) and bis(trimethylsilyl)sulfide (($TMS)_2S$). OT is well-known to grow highly crystalline and high QY QDs[5,42]. OT grows high-quality core-shell QDs from big cores (diameter ($d$): 4.6 nm, $T_m$: 250 °C; Fig. 2a, b, d). These QDs are highly crystalline (92%, 276/300) and have high QY and narrow full width at half maximum (FWHM) (25 nm). However, the high growth temperature (~300 °C) imposed by the low reactivity of OT limits the application of OT to cores with low $T_m$ (e.g., small CdSe QDs and PbS QDs). As shown in Fig. 2a–c, CdS deposition onto small CdSe cores ($d$: 1.8 nm, $T_m$: 150 °C) using OT yields QDs with non-uniform shape and size, as well as broad emission (FWHM: 37 nm; Supplementary Table S2). Furthermore, Fig. 2a, e shows that the QDs grown with OT exhibit significantly larger PL shift than those grown with ($TMS)_2S$. These results indicate that QD growth at a temperature higher than the $T_m$ of cores leads to

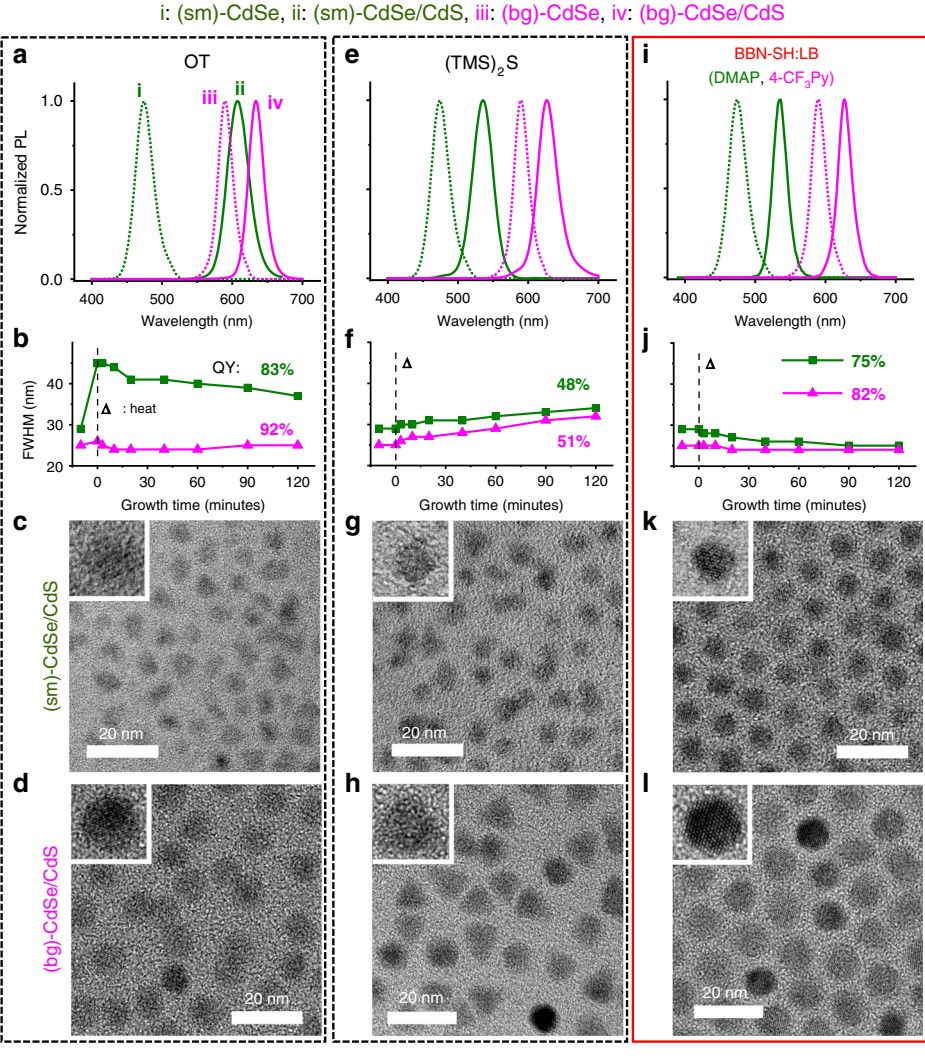

**Fig. 2 [BBN-SH:LB]-based high-quality shell growth in different sizes of QD cores.** [BBN-SH:LB] enables a systematic growth of high-quality shells while the QDs produced with conventional precursors show suboptimal quality for all ($(TMS)_2S$) or some (OT) of the sizes. **a**, **e**, **i** PL spectra of core (dotted line) and core-shell QDs (solid line). **b**, **f**, **j** Evolution of FWHM during growth (Δ: the reaction temperature was increased for each shell deposition process, see Supplementary Table S4 for detail condition). **c**, **d**, **g**, **h**, **k**, **l** TEM images (inset: HRTEM images) of (sm)-CdSe/CdS (**c**, **g**, **k**) or (bg)-CdSe/CdS core/shell QDs (**d**, **h**, **l** scale bar: 20 nm). When using BBN-SH, tight size distribution (FWHM < 25 nm) is obtained for both (sm)- and (bg)-CdSe cores, whereas OT produces QDs with narrow FWHM only from (bg)-CdSe cores. In the case of $(TMS)_2S$, the emission of the final QDs is broader (FWHM > 32 nm) for both small and big cores. The TEM images confirm narrow size distribution, shape uniformity, and high crystallinity for all size QDs produced with BBN-SH. The structural quality of these QDs is comparable to the QDs grown with OT from big cores.

particle disintegration[43] and undesired alloying[44] at the core-shell interface. Although $(TMS)_2S$ produces heterogeneous core-shell structure due to the low-temperature growth, the low reactivity of OT requires high-temperature growth and yields an alloyed interface. $(TMS)_2S$ is often used for low-temperature growth due to its high reactivity. However, even at very low temperatures (100–130 °C), its reactivity is still high, inducing uncontrolled growth[29,32]. As a result, the produced QDs have low crystallinity (45 (135/300)–61% (183/300)), broad emission (32–34 nm), non-spherical shape (circularity: <0.5), and low QY (Fig. 2e–h). To evaluate the crystallinity of each sample, we counted the percentage of particles showing clear fringes from more than 300 particles that are randomly selected (Supplementary Fig. S5)[45]. These examples emphasize the limitation of conventional precursors whose reactivity cannot be tuned. They either grow high-quality nanocrystals only at a narrow range of growth temperatures and from cores of specific sizes or fail to achieve optimal surface-reaction kinetics, generating suboptimal quality QDs.

**Systematic growth of high-quality shells onto cores of various sizes using BBN-SH.** Unlike conventional precursors, the reactivity of BBN-SH can be predictively tuned using chemical additives. We optimized the reaction condition by examining the impact of growth temperature and precursor reactivity to the structural quality of QDs. We found that the well-controlled growth of shells occurs at a temperature that is 10–20 °C lower than the $T_m$ of cores. At higher temperatures, surface atoms become excessively labile, leading to undesired alloying or dissolution of surface atoms (Supplementary Fig. S6). In comparison, lower temperatures induce less controlled and more faceted growth (Supplementary Fig. S7). Three reaction regimes exist for crystal growth as the reaction temperature varies. At the lowest temperature regime, the added precursor does not have enough energy for surface diffusion, producing particles with irregular shapes. At a higher-temperature regime, the added precursor diffuses to the optimal site inducing surface-reaction-limited growth that develops thermodynamically preferred facets (Supplementary Fig. S8, see the Supplementary

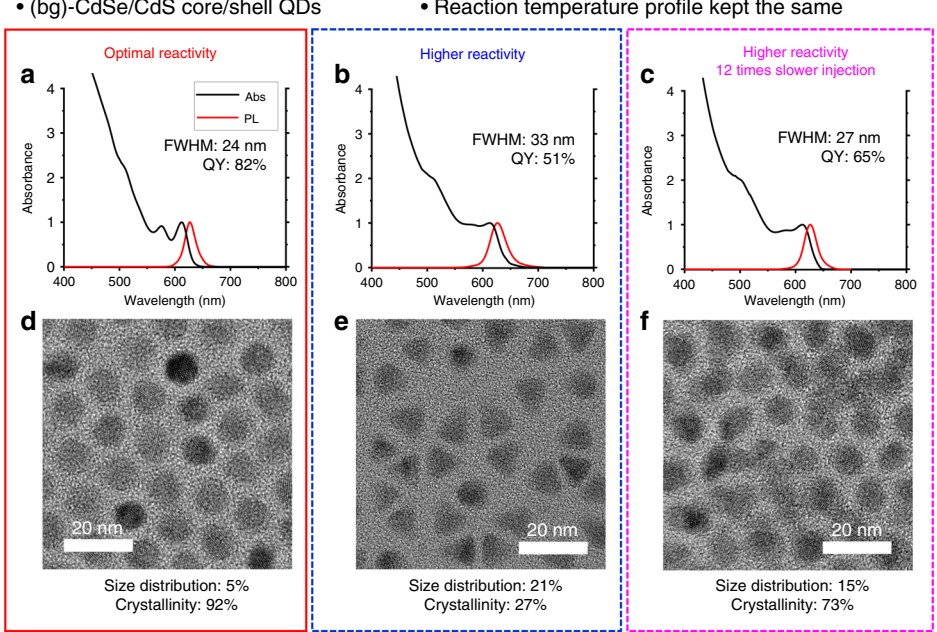

**Fig. 3 Impact of precursor reactivity to the optical and structural quality of QDs.** The (bg)-CdSe/CdS QDs grown at the same reaction temperature by infusing (**a–d**) [BBN-SH:4-CF$_3$Py] for 2 h, (**b**, **e**) [BBN-SH:DMAP] for 2 h, or (**c–f**) [BBN-SH:DMAP] for 24 h. **a–c** Absorbance and PL spectra of (bg)-CdSe/CdS QDs. **d–f** TEM images of (bg)-CdSe/CdS QDs (scale bar: 20 nm). Higher reactivity of the precursor yields lower QY, inferior crystallinity, broader FWHM, and uncontrolled growth. A significant reduction in the infusion rate partially improves the quality of the produced QDs.

Information for detail explanation and Supplementary Results). As the reaction temperature approaches the melting temperature of nanoparticles, the energy difference between facets decreases, creating spherical particles even with thermodynamically controlled growth[3,46,47]. In this study, we focus on the third regime, as spherical QDs are preferred in most optoelectronic applications with their superior optical and electronic properties. Precursor reactivity is a parameter that also carries a significant impact on the crystallinity of QDs, but it has been optimized only for size focusing. Precursor reactivity can be defined in various ways, but in our study, precursor reactivity is characterized by the temperature that initiates QD growth. Figure 3 compares QDs that are grown at the same temperature but with LBs yielding significantly different $T_{act}$ (4-CF$_3$Py vs. DMAP). In both cases, $T_{act}$ is lower than the growth temperature to ensure effective shell growth. The QDs grown with BBN-SH and 4-CF$_3$Py, which initiate shell growth at the higher temperature, exhibit remarkably higher crystallinity (92% vs. 73%), more uniform shape (circularity: 0.84 vs. 0.78), higher QY (82% vs. 65%), tighter size distribution (5% vs. 15%), and narrower FWHM (24 vs. 27 nm) (Fig. 3 and Supplementary Table S3). The negative impact of highly reactive precursors (BBN-SH with DMAP) can be partially compensated by slowing down the rate of shell growth; however, even with a 12-fold decrease in precursor injection rate, the quality of the final QDs are significantly worse than those produced using BBN-SH and 4-CF$_3$Py (Fig. 3c, f and Supplementary Fig. S9). This result emphasizes the difference between the strategies used for size focusing vs. quality improvement. For size focusing, the concentration of activated solutes is controlled by adjusting either the reactivity or infusion rate of precursors[34]; however, to improve the structural quality of nanocrystals, optimizing the precursor reactivity itself is crucial. QDs with the best quality are synthesized with an LB yielding $T_{act}$ that is 30 °C lower than the growth temperature (Supplementary Fig. S10). LBs yielding even lower $T_{act}$ produce QDs with non-uniform size and shape, lower QY, and significantly broad FWHM, whereas LBs that yield higher $T_{act}$ result in inefficient growth and lower QY. With appropriate LBs, BBN-SH allows high-quality shell growth a

wide range of temperatures (100–230 °C), covering most QDs of different sizes and materials. This temperature range can be further extended by using different LBs. Of note, a library of thiourea precursors allow shell growth at a limited range of temperature (60–110 °C), which is too low for efficient shell growth, and easily form satellite particles (Supplementary Fig. S11).

Using the optimized growth condition, we successfully produced high-quality CdSe/CdS QDs from CdSe cores of various sizes with a well-defined core-shell structure ($d$: 1.8–4.6 nm). Figure 2 presents the QDs produced from the smallest and biggest cores. Different from conventional precursors (Fig. 2a–h), BBN-SH produces QDs with high monodispersity (<5% deviation), narrow emission (FWHM: <25 nm), high crystallinity (92–95%), spherical shape (circularity: >0.8), and high QYs (87–91%) from all size cores (Fig. 2i–l, Supplementary Fig. S12, and Supplementary Table S2). It is notable that these QDs have comparable quality to the QDs created using OT and big cores, a method known to create the highest quality QDs. This result shows that after optimization of the growth condition, high-quality QDs can be produced in a mild condition. In comparison, OT-based growth requires high growth temperature (~310 °C) due to its low reactivity[5]. Of note, the QDs grown with BBN-SH and pyridine-based LB are post-treated with CdCl$_2$ to mitigate the adsorption of pyridine-based byproducts, which is known to induce PL quenching[48]. Without CdCl$_2$ treatment, the QY is 75-82%. In comparison, the QDs grown with OT do not show any increase in their QY after CdCl$_2$ treatment (91→91%) and the QDs grown with (TMS)$_2$S show only a marginal increase (51→53%; Supplementary Table S2). The slight quenching due to pyridine-based byproduct can be avoided by using non-pyridine-based LB.

To characterize the crystallinity of QDs, we used various characterization techniques (Supplementary Table S5). First, we analyzed TEM images to examine the fringe pattern of each nanocrystal as described above. Examining TEM images allows visual inspection of the quality of particles, but only a small fraction of particles is analyzed, and the fringe patterns are only

observed from crystalline particles having specific orientations. To complement the limitations of TEM analyses, we analyzed the diffraction pattern of QD samples from X-ray diffraction (XRD) and selected area electron diffraction (SAED). The FWHM of diffraction peaks is positively correlated to the density of defects. To eliminate the influence of stacking faults to the broadness of diffraction peaks, we simulated the XRD profile of CdS nanoparticles having various levels of stacking faults and identified a peak that is not affected by the level of stacking faults (details provided in Supplementary Information). As shown in Supplementary Fig. S13, the (11$\bar{2}$0) peak is not affected by stacking faults; therefore, it is used for this analysis. Both XRD and SAED results confirm the TEM analysis: the QDs produced with OT and the optimal pair of BBN-SH and LB show similar peak broadness, while the QDs produced with the unoptimized pair of BBN-SH and LB show the significantly broader peak (Supplementary Fig. S9). Finally, we analyzed the single dot blinking statistics of different QD samples to compare their structural quality. Structural defects often serve as midgap states, impacting the photophysics of QDs and promoting the non-radiative decay. Thus, the blinking statistics serve as an indirect measure of structural quality. Of note, the photophysics of QDs is strongly affected by the core/shell interface structure (e.g., heterostructure vs. gradient shell)[27,28]. Therefore, only the QDs having the same core/shell interface structure can be compared. Here, we compared the average ON-time fraction ($\Phi_{on}$), the average fraction of time that each QD stays ON during the measurement, of the QDs grown with the optimal LB (4-CF$_3$Py) vs. those grown with the suboptimal LB (DMAP). To extract $\Phi_{on}$, more than 50 individual blinking traces per sample, each from a different QD, are analyzed. Supplementary Fig. S14 shows that the $\Phi_{on}$ is significantly larger for the QDs grown with the optimal LB (0.81 vs. 0.62). These results suggest that the QDs grown with the optimal LB have fewer defects than the QDs grown with the suboptimal LB and non-radiative decays are more prominent in the unoptimized sample. These analyses show that the crystallinity of all QDs created with

BBN-SH is comparable with big QDs grown with OT (Supplementary Fig. S9). In comparison, conventional precursors either grow high-quality shells only from cores of limited sizes or produce low quality shells, even upon extensive optimization of growth condition (Fig. 2a–h).

**Detailed QD growth process and its impact on the crystal quality.** Close examination of the growth results provides insight into a nanoparticle growth process. Figure 4 illustrates detailed steps for QD growth. Upon injection, precursors are converted into activated solutes. These solutes are then adsorbed to the QD surface (i). Concurrently, the thermal energy of nanoparticles causes surface atoms to be desorbed (ii). The adsorbed atoms on the surface are then activated; this step often involves ligand cleavage (iii). The activation energy for this process ($E_{a3}$) is dependent on the reactivity of precursors. The activated atoms then undergo surface migration and relaxation (iv). To generate a defect-free and highly crystalline surface, the mean-squared diffusion length of adatoms must be long enough so that they settle at an optimal location and configuration[49]. The surface reconfiguration process is quenched upon the collision with other adatoms; this undesired process results in uncontrolled growth (v). Table 1 summarizes the results of suboptimal reaction conditions for shell growth. Excessive heating causes surface desorption to dominate over adsorption, inducing undesired alloying or Ostwald ripening (Fig. 2a, c). High reactivity precursors yield a high density of activated adatoms promoting the undesired quenching of surface relaxation, therefore producing QDs with irregular

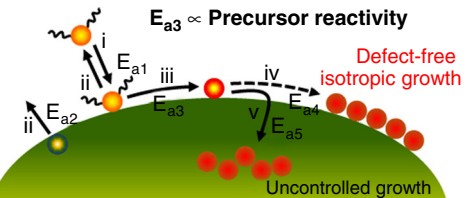

**Fig. 4 Detailed processes involved in QD growth.** Activated solutes are adsorbed to the surface (i), while the thermal energy of the surface causes surface atoms to be desorbed (ii). The adsorbed atoms are then activated (iii), followed by surface migration and relaxation. The mean-squared diffusion length of adatoms must be long enough to induce defect-free isotropic growth (iv). If the density of activated adatoms is excessive, collision between adatoms quenches surface relaxation, leading to uncontrolled growth (v). The activation energy of step iii ($E_{a3}$) is mainly affected by the reactivity of precursors. In comparison, temperature is the key parameter affecting the activation energy of other steps.

**Table 1 Summary of the growth results due to suboptimal reaction conditions.**

| Low growth temperature | High growth temperature |
|---|---|
| Low precursor reactivity | |
| Inefficient shell growth | Undesired alloying particle disintegration |
| High precursor reactivity | |
| Uncontrolled shell growth | Optimal for nucleation not for shell growth |

shape (Fig. 3). Slow infusion of precursors compensates for this negative impact by reducing the density of activated adatoms on the surface; however, insufficient precursor supply causes surface desorption to dominate over adsorption, inducing unstable growth (Fig. 3c, f).

Precursors with insufficient reactivity also produce QDs with reduced crystallinity (Supplementary Fig. S10). This is because inefficient growth increases the minimum diffusion length required for defect-free growth, which promotes incomplete surface relaxation. Detailed understanding of the growth process allows a systematic optimization of reaction conditions to produce high-quality nanocrystals of various sizes and materials, including non-sulfur-containing QDs and other types of inorganic nanoparticles.

**Systematic growth of high-quality core-only QDs using BBN-SH.** The growth principles established above apply to both core-shell and core-only QDs. We demonstrate that we can also grow high-quality cores of various materials using our precursor chemistry. To decouple growth from nucleation, we first generated small nuclei, removed unreacted precursors, and initiated growth by adding fresh precursors, including BBN-SH and LB. We performed the growth studies with lead sulfide (PbS) and copper indium sulfide (CuInS$_2$) QDs. PbS QDs are promising materials for flexible and low-cost optoelectronics. Due to highly labile surface atoms, the $T_m$ of PbS QDs is particularly low (40–150 °C)[41]. Low growth temperature restricts the choice of precursors to highly reactive ones, which often leads to poor crystal quality (Supplementary Fig. S15)[50]. A recent study reports that monodisperse PbS QDs with high crystallinity can be prepared by using elemental sulfur and PbCl$_2$, whereas the lead precursor being in large excess (Pb:S = 8–24:1)[51,52]. In this method, however, >97% of Pb precursors are left unconsumed after QD growth. The multi-step, expensive, overnight purification scheme required for this method limits their applications to

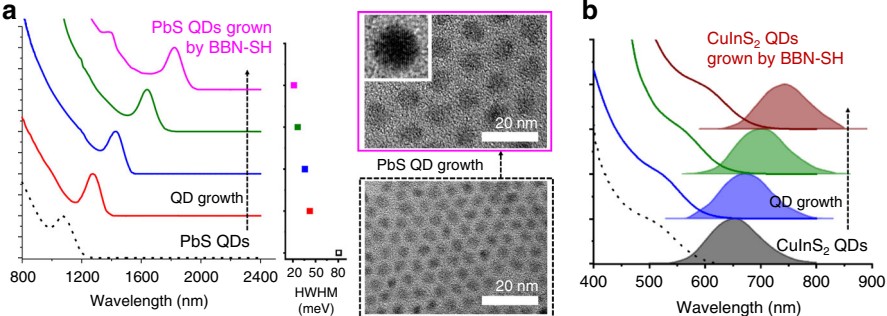

**Fig. 5 Successful growth of high-quality PbS and CuInS$_2$ QDs using BBN-SH. a** Absorbance spectra of PbS QDs during growth. HWHM (meV) progressively narrows during growth (PbS nuclei: black, PbS QDs grown by BBN-SH: other colors). The TEM images of the final QDs and PbS nuclei. (inset: HRTEM image; scale bar: 20 nm). **b** Absorbance and PL spectra of CuInS$_2$ QDs during growth (CuInS$_2$ nuclei: black; CuInS$_2$ QDs grown by BBN-SH: other colors).

large scale synthesis and commercial products. Our precursor chemistry overcomes this limitation with its capability to efficiently grow high-quality PbS QDs with full conversion of precursors (Fig. 5a). Upon the addition of growth precursors (Pb (oleate)$_2$:BBN-SH:DMAP = 1:1:1), the half width at half maximum (HWHM) of the first absorption peak decreases significantly (81–42 meV). The HWHM progressively decreases as PbS QDs grow. After growing 3 MLs of PbS, the size distribution of the ensemble becomes close to the homogeneous limit (HWHM: 21 meV). The produced QDs also show high crystallinity, evidenced by clear fringes in TEM images and narrow XRD peaks (Supplementary Fig. S16). Due to the full conversion of precursors, the conventional and simple crash-out method is used for purification, greatly simplifying the post-synthesis processing. It is noteworthy that such quality PbS QDs has not been achieved using the method yielding the full conversion of precursors. We also demonstrate the growth of high-quality CuInS$_2$ QDs using our precursor chemistry. CuInS$_2$ QDs has been recognized as an environmentally-friendly alternative to Cd- or Pb-based QDs with their band gap in the near-infrared region, a large absorption coefficient, and high photostability[53,54]. Figure 5b shows a controllable growth of CuInS$_2$ using BBN-SH and 3-ClPy. Supplementary Fig. S17 shows that the produced CuInS$_2$ QDs show clear fringes, indicating high-quality crystals. Elemental analysis using inductively coupled plasma-optical emission spectrometry confirmed the ratio of Cu:In:S remains 1:1:2 after the QD growth (Supplementary Table S6). A relatively broad emission is mainly attributed to the random positioning of the Cu-related defect emitting center[55]. These results emphasize the wide applicability of the BBN-SH chemistry to any sulfur-containing QDs.

## Discussion

We have established a systematic approach to grow high structural quality QDs. In contrast to size control, the crystallinity and morphology of QDs have been improved mostly by trial and error or a material-specific manner. For old systems, such as CdSe/CdS QDs, their synthesis has been extensively optimized to yield QDs having almost unity QY and high crystallinity[5,56]. Common approaches include varying ligands[56], a temperature profile during reaction[5], and precursors[35], but the established strategies are only applicable to the QDs of a specific material or size range. Here we identify the key parameters impacting the quality of nanocrystals, including their crystallinity, morphology, density of defects, and core-shell interface structure, and introduce a design principle for precursors that enables systematic optimization of the growth condition. We show that precursor reactivity and growth temperature govern the structural quality of nanocrystals. Growth temperature governs how labile the surface atoms are,

whereas precursor reactivity affects the reaction kinetics on the QD surface. These two parameters must be tuned in a systematic manner to create highly crystalline, spherical QDs with minimal defects and a well-controlled core-shell structure. Systematic variation of precursor reactivity is challenging with existing precursors. The reactivity of precursors is mostly fixed by their chemical structure, requiring the synthesis of different molecules for reactivity variation. Also, in most cases, the existing precursor selections only covers a narrow range of reactivity, failing to induce controlled growth for QDs of different sizes and materials. To address this limitation, we designed a sulfur precursor, BBN-SH, whose reactivity can be chemically tuned in a predictable manner. With LB coordination, BBN-SH covers a broad range of reactivity, allowing their applications to QDs of different materials and sizes. By monitoring the structural quality of the QDs while varying both growth temperature and precursor reactivity, we identified the universal guideline for growth conditions that can be applied to QDs of different sizes and materials. The detailed understanding of the growth process allows logical tuning of the growth condition. We show that BBN-SH can be used as a universal precursor to systematically grow high-quality QDs of various sizes, materials (CdS, PbS, and CuInS$_2$), and types (core vs. core/shell); resulting QDs have high crystallinity, spherical shape, narrow emission, high QY, excellent monodispersity, and a well-defined core-shell interface structure. Our result emphasizes the benefit of employing a chemically modulatable bond in nanoparticle precursors. The QD growth principle established here inspires the design of anion precursors for other types of nanoparticles, including metal chalcogenides and pnictogenides QDs. This study provides a foundation for the controlled growth of high-quality nanocrystals.

## Methods

**Initiation CdS shell growth by picoline and BBN-SH.** CdSe cores (50 nmol; medium CdSe core (535 nm absorption peak)) were isolated by repeated precipitations from hexane with acetone. The CdSe QD cores were redispersed in a minimal amount of hexane and loaded in a solvent mixture of 3 mL of OAm and 3 mL of octadecene (ODE). The reaction solution was degassed under vacuum at 100 °C for 1 h, then the temperature was dropped to 60 °C. For the first round of CdS shell growth, the amount of each Cd or S precursor was calculated for growing 5 ML of CdS shell on the CdSe QD cores. The Cd precursor (0.15 mmol Cd-oleate in ODE) and the S precursor (0.15 mmol of BBN-SH in 1 mL of OAm, and 1 mL of TOP) were injected to the CdSe core solution sequentially. The reaction temperature for the mixture of CdSe cores and shell precursors was heated to 170 °C for 20 min. Picoline (0.15 mmol) in 1 mL of ODE was injected for 20 min and CdS shell growth was initiated. The reaction temperature was increased by 15 °C every 5 min up to 200 °C. The reaction temperature was kept at 200 °C for another 110 min. The temperature was then dropped to 60 °C quickly. For the second round of CdS shell growth, the amount of each Cd or S precursor was calculated for growing additional 2 ML of CdS shell layers. The Cd precursor (0.13 mmol Cd-oleate in ODE) and the S precursor (0.13 mmol of BBN-SH in 1 mL of OAm, and 1 mL of TOP) were injected to the CdSe core solution sequentially. The mixture of CdSe

cores and shell precursors was heated at 200 °C for 20 min. Picoline (0.13 mmol) in 1 mL of ODE was injected for 10 min and a second round of CdS shell growth was initiated. The reaction temperature was kept at 200 °C for another 80 min. After the CdS growth, the reaction temperature was dropped to room temperature quickly.

**Characterization of the $T_{act}$ upon the addition of different LBs.** CdSe cores (50 nmol) were isolated by repeated precipitations from hexane with acetone. The CdSe QD cores were redispersed in a minimal amount of hexane and loaded in a solvent mixture of 3 mL of OAm and 3 mL of ODE. The reaction solution was degassed under vacuum at 100 °C for 1 h and dropped to 60 °C. The amount of each Cd or S precursor was calculated for growing 4 ML of CdS shell on the CdSe QD cores. The Cd precursor (0.11 mmol Cd-oleate in ODE) and the S precursor (0.11 mmol of BBN-SH in 1 mL of OAm and 1 mL of TOP) were injected to the CdSe core solution sequentially. Each chemical initiator (0.11 mmol; DMAP, picoline, 3-ClPy, 4-CyPy, 4-(trifluoromethyl)pyridine, 1-methylpyrrolidine, DABCO, and 1-phenylimidazole) was added in the mixture. To determine the $T_{act}$ by different types of activators, the reaction temperature for the mixture of CdSe cores and shell precursors was slowly increased by 10 °C over 20 min and maintained for another 20 min. For thioureas, either $N,N'$-diphenylthiourea or $N$-$n$-hexyl-$N'$-dodecylthiourea was used as sulfur precursor. The total amount of each Cd or S precursor was calculated for growing 4 ML of CdS shell on the CdSe QD cores. While the temperature profile was kept the same as above, the amount of Cd or S precursors for growing 0.5 ML of CdS shell were added sequentially (in 10 min gap between Cd and S precursor injection) every step of 10 °C increasing in reaction temperature.

## Data availability
The authors declare that the data supporting the findings of this study are available within the paper and its Supplementary Information files.

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

## Acknowledgements

This work was supported by a start-up fund from the University of Illinois at Urbana-Champaign. G.W.H. acknowledges the financial support from the Korea Institute of Science and Technology (KIST) Institution Program (Grant number 2E30100). We thank Professor Thomas B. Rauchfuss for providing helpful discussions. We thank the SCS NMR Lab and George L. Clark X-Ray Facilities, University of Illinois, for its technical support. The Varian Inova 400-MHz NMR spectrometer was obtained with the financial support of the Roy J. Carver Charitable Trust. Transmission electron microscope measurements were carried out in part in the Materials Research Laboratory Central Research Facilities, University of Illinois. Single QD blinking traces were acquired at Carl R. Woese Institute for Genomic Biology on a demo Alba system provided by ISS, Inc. We thank ISS, Inc. for providing this instrument and Austin Cyphersmith for training and assistance using the Alba system.

## Author contributions

J.P. and H.-.S.H. conceived the project and designed the experiments. J.P. and A.J. performed all of the experiments and measurements. A.W.S. performed the single QD blinking measurements. G.W.H. performed XRD simulations. J.P., G.W.H., and H.-S.H. analyzed the data. All the authors contributed to the manuscript preparation.

## Competing interests

The authors declare no competing interests.
