## [Peer Review File · Nature Communications]

REVIEWER COMMENTS

Reviewer #1 (Remarks to the Author):

Development of a systematic way to control nanocrystal(NC) growth pose huge impacts. Till now, all the precursors used for NC syntheses have relied on thermally-induced precursor conversion (or activation) to a form they can be readily incorporated into NC lattices. For an example, a tunable library of substituted thiourea precursors has been recently reported to control the thermal reactivity of NC precursors (Science, 2015, 348, 1226-1230. Ref. 19 in this manuscript). Authors are reporting chemically-induced precursor conversion using a family of sulfur precursors bearing a boron-sulfur bond that can be temperature-independently activated by different Lewis bases. This reviewer believes that this conceptual advance of 'chemically-induced precursor conversion for NC growth' meets the high standard and well merits the publication in Nature Communications. The concept reported herein and the design could be broadly applied for systematic syntheses for more elegant NCs that can outperform for many applications. However, a few issues need to be properly addressed as listed below.

1. Authors have demonstrated synthesis of high-quality wurzite CdSe/CdS QDs using their chemically-inducible precursors. Thiols and (TMS)₂S reagents were used as control precursors that yielded suboptimal QDs. For zinc-blende CdSe/CdS QDs, the synthesis has been already quite advanced. This manuscript should be a proof-of-concept that this kind of new precursors can be advantageous for developing new NCs. For an old system like CdSe/CdS QDs, the synthesis has been really extensively optimized using old precursors by many researchers which includes usage of multiple ligands, temperature profile control over reaction time, and so forth (J. Am. Chem. Soc. 2017, 139, 16556-16567). Authors are not reporting 'champion' QDs nor to be the proper purpose of this manuscript. Discussion in this regard should be properly expanded.
2. In the same vein as comment 1, more rigorous comparison should be made between samples synthesized by chemically-inducible precursors Vs. controls by conventional precursors. Authors mostly rely on ensemble optical properties (PL wavelength and band width) and TEM images. Crystallinity comparison by TEM image is quite limited. Although authors provided XRD data, the analyses were rather poor. The XRD data should be further analyzed for wurzite/zinc-blende stacking faults (ABCABC Vs. ABABAB). XRD simulations may help such analyses.
3. Single QD emission studies should be included. Blinking dynamics directly reflects the individual environment of QDs and can be a good standard assessing the quality of QDs.
4. This reviewer does not fully understand why authors stick to spherical QDs. Their new precursors allow temperature independent activations which may be more suited for syntheses of anisotropic NCs. Authors need to compare the shape of NCs by their precursors at different temperatures. For example, shapes of NCs could be compared for the four suboptimal categories in Table 1.

Below are minor comments.

1. On page 3, authors claimed "On the other hand, high temperature growth is preferred to promote thermal annealing of defects and kinetically controlled growth, which creates spherical QDs". Isn't it typically "thermodynamically controlled growth" at high temperature?
2. FWHM data of small CdSe cores in Figure 2b are strange. The three sets of experiments start with identical CdSe cores. The FWHMs should be same at 0 minutes for the three sets (b, f, and j in Figure 2).

3. In Figure 2i-j, LBs used should be specified.

Reviewer #2 (Remarks to the Author):

In the submitted manuscript, Park et al. introduce a new sulfur precursor, whose reactivity can be systematically controlled for the growth of metal chalcogenide QDs. By employing R-B-SH precursors, the authors were able to obtain high-quality QDs at relatively mild reaction temperature. Given the potentially extensive applicability of the precursors, I believe this work is publishable in Nature Communications, if several minor issues are properly addressed.

1. It is clear that the optimized precursor and reaction temperature yields high quality CdSe/CdS QDs with high PL QY and narrow emission linewidth. However, the discussions of crystallinity base on inadequate characterizations and speculative arguments. Especially, the slight difference in FWHM of peaks in XRD patterns (Figure S7) does not solely reflect the concentration of defects. The samples in comparison do not have same shape, size, surface configuration, film thickness and roughness.
2. Second question on crystallinity discussion is the defining the crystallinity by authors. Authors defined 'crystallinity' to 'the number of particles show clear fringes over the number of particles in TEM images'. How the authors define the clearness of fringes? In addition, the fringes in TEM images are highly affected by the focusing of electron beam to sample in the microscope. Moreover, the extremely low portion of particles is not likely to represent the overall crystallinity of ensemble.
3. The discussion on the balance of surface instability and precursor supply is well written and corresponds to the experimental results. However, I consider that the discussion on the facet development during the growth is insufficient and somewhat speculative. I recommend to add more discussion in terms of growth regimes (diffusion or reaction limited).
4. Though the authors introduced the synthesis of CuInS₂ NCs using R-B-SH precursors, the main data consist of Cd and Pb based materials. I recommend to emphasize the extensibility of the authors' approach to other semiconductor NCs, such as metal selenides or pnictides.

<Our response to each reviewer #1's comments (NCOMMS-20-12630-T)>

Reviewer #1:

Development of a systematic way to control nanocrystal(NC) growth pose huge impacts. Till now, all the precursors used for NC syntheses have relied on thermally-induced precursor conversion (or activation) to a form they can be readily incorporated into NC lattices. For an example, a tunable library of substituted thiourea precursors has been recently reported to control the thermal reactivity of NC precursors (Science, 2015, 348, 1226-1230. Ref. 19 in this manuscript). Authors are reporting chemically-induced precursor conversion using a family of sulfur precursors bearing a boron-sulfur bond that can be temperature-independently activated by different Lewis bases. This reviewer believes that this conceptual advance of 'chemically-induced precursor conversion for NC growth' meets the high standard and well merits the publication in Nature Communications. The concept reported herein and the design could be broadly applied for systematic syntheses for more elegant NCs that can outperform for many applications. However, a few issues need to be properly addressed as listed below.

Thank you very much for your positive comments. We will address all your questions below.

Q1. Authors have demonstrated synthesis of high-quality wurzite CdSe/CdS QDs using their chemically-inducible precursors. Thiols and (TMS)₂S reagents were used as control precursors that yielded suboptimal QDs. For zinc-blende CdSe/CdS QDs, the synthesis has been already quite advanced. This manuscript should be a proof-of-concept that this kind of new precursors can be advantageous for developing new NCs. For an old system like CdSe/CdS QDs, the synthesis has been really extensively optimized using old precursors by many researchers which includes usage of multiple ligands, temperature profile control over reaction time, and so forth (J. Am. Chem. Soc. 2017, 139, 16556-16567). Authors are not reporting 'champion' QDs nor to be the proper purpose of this manuscript. Discussion in this regard should be properly expanded.

As the reviewer pointed out, we are not reporting 'champion' QDs. Rather, we reveal the main parameters contributing to the crystal quality of quantum dots (QDs) and report new precursor chemistry that enables systematic optimization of the growth condition to grow highly crystalline QDs. We have updated the manuscript to emphasize the importance of identifying a systematic method to grow high-quality crystals and the strength of our systematic approach over previous approaches. We also cited the previous works properly in the revised manuscript. Moreover, using the new precursor chemistry and optimized growth conditions, we produced both core-only and core/shell QDs having comparable qualities to the best quality QDs reported so far. We have included new data comparing the QY of the newly synthesized QDs versus the best quality QDs reported so far. We would like to emphasize that we created one of the best quality QDs using a **universal** precursor (without having to synthesize different precursors for different materials or sizes) and rather **mild** conditions in a **systematic** manner. We also updated the discussion section to emphasize that the BBN-SH chemistry provides a critical insight for new precursors for other QDs.

Significance of our work: Obtaining QDs with high crystal quality is crucial for various applications because structural defects often serve as electronic defects, reducing carrier mobility, and promoting the non-radiative decay process. So far, **no systematic method existed to grow high-quality crystals**. As mentioned by the reviewer, the ligands and temperature profile used for QD synthesis has been extensively optimized to obtain monodisperse QDs with high QY. However, most optimization methods to produce QDs with high QY are trial-and-error based. Instead of tuning the reaction parameters to "improve" the QY, most previous studies "report" the reaction conditions yielding high QY. Also, QY is highly related but not a direct measure of the crystal quality. Here, we examine a detailed surface reaction

involved for growth and show that systematic optimization of reaction temperature and precursor reactivity leads to the controlled growth of high-quality crystals.

Unique approach we are taking and its significance: Reaction temperature governs the surface instability of nanoparticles, while precursor reactivity regulates the reaction kinetics on the surface. High-quality crystal growth requires an optimal degree of surface instability and precursor reactivity, thereby, requesting for independent control of the two parameters. Unfortunately, all the existing precursors fail to modulate their reactivity independent of temperature. Recently, a few precursor libraries have been reported that modulate the reactivity of precursors by using varying the substituents. However, synthesizing different precursors is highly costly, labor-intensive, and low throughput. We report a new precursor, whose reactivity can be modulated in situ by adding Lewis base molecules having varying levels of basicity. We show that the new precursor grows core-only and core/shell QDs of various materials (CdSe/CdS, PbS, CuInS₂ QDs) and sizes and produces highly crystalline QDs. In contrast, the conventional precursors grow highly crystalline QDs only in limited cases or induce undesirable alloying.

The optical quality of new QDs being comparable to the best QDs reported so far: **Even though the claiming the ‘champion’ QDs was not our goal, the new precursor produced QDs having the QY close to unity (~ 90 %) after post-treatment with CdCl₂.** Post-treatment is required since the pyridine-based byproducts produced after precursor conversion can adsorb to the surface and quench the PL as described in the previous studies.¹ To confirm the potential quenching by the pyridine-based byproducts, we treated all QDs after growth with CdCl₂, including those grown with the conventional precursors and the new precursors. As expected, CdCl₂ treatment greatly increased the QY for the QDs grown with BBN-SH while only marginal improvement is observed for the QDs grown with the conventional precursors. We have added the new data and discussion in the manuscript as below.

(Page 10) Different from conventional precursors (Figure 2a-h), BBN-SH produces QDs with high monodispersity (<5% deviation), narrow emission (FWHM: < 25 nm), high crystallinity (92-95%), spherical shape (circularity: > 0.8), and high QYs (87-91%) from all size cores (Figure 2i-l, S12, Table S2). It is notable that these QDs have comparable quality to the QDs created using OT and big cores, a method known to create the highest quality QDs. This result shows that after optimization of the growth condition, high quality QDs can be produced at a mild condition. Of note, the QDs grown with BBN-SH and pyridine-based LB are post-treated with CdCl₂ to mitigate the adsorption of pyridine-based byproducts, which is known to induce PL quenching.⁴³ Without CdCl₂ treatment, the QY is 75-82%. In comparison, the QDs grown with OT do not show any increase in their QY after CdCl₂ treatment (91→91%) and the QDs has grown with (TMS)₂S show only a marginal increase (51→53%, Table S2). The slight quenching due to pyridine-based byproducts can be avoided by using non-pyridine-based Lewis base.

(Page 14) We have established a systematic approach to achieve high structural quality QDs. In contrast to size control, the crystallinity and morphology of QDs have been improved mostly by trial and error. For old systems, such as CdSe/CdS QDs, their synthesis has been extensively optimized to yield QDs having almost unity QY and high crystallinity.^{5, 55} Common approaches include varying ligands⁵⁵, a temperature profile during reaction⁵, and precursors³⁵, but the established strategies are only applicable to the QDs of a specific material or size range. Here, we identify the key parameters impacting the quality of nanocrystals, including their crystallinity, morphology, density of defects, and core-shell interface structure, and introduce a new design principle for precursors that enables systematic optimization of the growth condition. We show that precursor reactivity and growth temperature govern the structural quality of nanocrystals.

(Page 15) Our result emphasizes the benefit of employing a chemically modulatable bond in nanoparticle precursors. The QD growth principle established here inspires the design of new anion precursors for other types of nanoparticles, including metal chalcogenides and pnictogenides QDs. This study provides a foundation for the controlled growth of high quality nanocrystals.

2. *In the same vein as comment 1, more rigorous comparison should be made between samples synthesized by chemically-inducible precursors Vs. controls by conventional precursors. Authors mostly rely on ensemble optical properties (PL wavelength and band width) and TEM images. Crystallinity comparison by TEM image is quite limited. Although authors provided XRD data, the analyses were rather poor. The XRD data should be further analyzed for wurzite/zinc-blende stacking faults (ABCABC Vs. ABABAB). XRD simulations may help such analyses.*

Thank you very much for your question. We recognize that the crystallinity assessment of QDs using TEM images and ensemble optical properties alone may not be sufficient. We added additional analysis and simulation data for the XRD results to supplement our statement. We also added new data on the selected area diffraction (SAED) pattern analysis using TEM, a technique that is used to assess the quality of crystals. Approximately 1000 particles are imaged to obtain the SAED patterns. Due to the random orientation of particles, ring patterns were observed rather than a series of spots. The linewidth of SAED rings is proportional to the FWHM of XRD peaks and is not affected by the thickness, roughness, and surface morphology of films. After integrating along the radial direction, we compared the FWHM of the (11 $\bar{2}$ 0) peak. Of note, this peak is not affected by the degree of stacking fault. Consistent with our XRD data, the FWHM of this peak is the largest for the mismatched BBN-SH:LB pair. The QDs prepared with octanethiol and the optimized BBN-SH:LB pair show almost identical FWHM, which is significantly smaller than the mismatched pair sample. This result reaffirms our conclusion that upon optimization, BBN-SH grows comparable quality QDs as octanethiol, the precursor known to produce the best quality QDs so far. We added the new data in Figure S13 and 14. Moreover, we would like to note that TEM-based analysis of the crystallinity of QDs is performed on a large number of nanoparticles (> 300) using a computer-aided and semi-automated image processing method. The large number ensures unbiased profiling and the computer-aided image processing method reduces the ambiguity in interpretation. To indicate the number of particles examined from TEM images, we added the number of crystalline particles and the total number of particles counted next to the percentage crystallinity data.

Further analysis and new simulation data for the interpretation of the XRD data: We would like to first note that the coherent volume in QDs is tiny compared to that of the bulk due to their small size. As a result, the XRD peaks QDs are significantly broader than that of the bulk. Broad peaks and long peak tails induce significant overlaps between peaks, making it difficult to extract the FWHM of each peak. With this limitation, we chose the peak at 43° corresponding to the (11 $\bar{2}$ 0) plane of wurzite CdS for each QD sample, and compared their FWHM. Furthermore, we calculated the size of coherent volume using the Debye-Scherrer's equation for each sample. This data is added to Figure S9b. Additionally, we performed simulations on the influence of stacking faults on the broadness of the peaks. We simulated XRD powder patterns using DIFFaX v1.812. Figure S14b shows the bulk result. As the stacking fault increases, the peak at 28.2° progressively broadened. However, in our nanoparticle XRD patterns, this peak significantly overlaps with the neighboring peaks, making it difficult to extract its FWHM. For the peak at 43°, the broadening is not observed. Therefore, we can conclude the broadening of the 43° peak is not due to the stacking fault and rather due to the other crystalline defects, such as antisite defects and dislocations. Detailed discussion and data are added in the main text and SI.

Figure S9. Higher precursor reactivity negatively impacts the crystallinity of QDs. (a) XRD data and (b) the summary of the CdS wurtzite(wz) (11 $\bar{2}$ 0) peak (“i”) and the size of coherent volume from (bg)-CdSe/CdS core/shell QDs synthesized by either octanethiol (olive), [BBN-SH:4-CF₃Py] pair (red), or [BBN-SH:DMAP] pair with 12 times slower injection (magenta).

Figure S13. Higher precursor reactivity negatively impacts the crystallinity of QDs. (a-c) Selected area electron diffraction (SAED) pattern (scale bar: 10 (1/nm)), (d) the projected 1D profile along the radius of diffraction ring pattern for resulted QD samples, (e) the FWHM of the arc from the CdS wurtzite(wz) (11 $\bar{2}$ 0) peak (“i”) from (bg)-CdSe/CdS core/shell QDs synthesized

by either octanethiol (olive), [BBN-SH:4-CF₃Py] pair (red), or [BBN-SH:DMAP] pair with 12 times slower injection (magenta), and (f-h) the projected 1D profile after baseline correction and peaks with multiple Gaussian fitting.

Figure S14. Stacking fault does not affect on the FWHM of the CdS wurtzite(wz) ($11\bar{2}0$) peak. Simulated XRD data from (a) nanocrystals or (b) bulk comprised with CdS wurtzite crystal (host) and zinc-blende (defect). FWHM of ($11\bar{2}0$) peak (“i”) in simulated XRD data for both nanocrystal (1.4) and bulk (0.1) was maintained while varying the percentage of stacking faults.

(Page 10) To characterize the crystallinity of QDs, we used various characterization techniques (Table S5). Firstly, we analyzed TEM images to examine the fringe pattern of each nanocrystal as described above. Examining TEM images allows visual inspection of the quality of particles but only a small fraction of particles is analyzed. To complement the limitations of TEM analyses, we analyzed the diffraction pattern of QD samples from X-ray diffraction (XRD) and selected area electron diffraction (SAED). The FWHM of diffraction peaks is positively correlated to the density of defects. To eliminate the influence of stacking faults to the peak broadness, we simulated the XRD profile of CdS nanoparticles having various levels of stacking faults and identified a peak that is not affected by the level of stacking faults (details provided in SI). As shown in Figure S14, the ($11\bar{2}0$) peak is not affected by stacking faults; therefore, it is used for this analysis. Both XRD and SAED results confirm the TEM analysis: the QDs produced with OT and the optimal pair of BBN-SH and LB show similar peak broadness, while the QDs produced with the unoptimized pair of BBN-SH and LB show the significantly broader peak (Figure S9, S13).

3. *Single QD emission studies should be included. Blinking dynamics directly reflects the individual environment of QDs and can be a good standard assessing the quality of QDs.*

We agree with the reviewer that single dot blinking statistics can be an additional measure for the crystal quality of QDs. We have included the representative blinking trace and the histogram of ON-fractions for the QDs prepared with BBN-SH (Figure S15). We only compare optimal and unoptimal pair ([BBN-SH:4-CF₃Py] and [BBN-SH:DMAP]), which have the same core-shell interface structure (hetero core-shell structure instead of gradient shells). The photophysics of gradient shell QDs cannot be directly compared to the QDs having heterostructures because a gradient shell creates a smooth potential well, greatly suppressing Auger recombination.^{2,3} The comparison shows that the average ON-time fraction of resulted QDs from optimal pair is much larger than that from unoptimal pair (0.81 vs 0.62). These results confirmed the crystallinity of the optimal pair sample is higher than that of the unoptimal pair sample, which also corresponds with XRD and SAED data. We summarized the all data and added in Table S5.

(Page 11) Finally, we analyzed the single dot blinking statistics of different QD samples to compare their structural quality. Structural defects often serve as midgap states, impacting the photophysics of QDs and promoting the non-radiative decay. Thus, the blinking statistics serve as an indirect measure of structural quality. Of note, the photophysics of QDs is strongly affected by the core/shell interface structure (e.g. heterostructure vs gradient shell)^{27, 28}, therefore, only the QDs having the same core/shell interface structure can be compared. Here, we compared the average ON-time fraction (Φ_{on}), the average fraction of time that each QD stays ON during the measurement, of the QDs grown with the optimal LB (4-CF₃Py) versus the suboptimal LB (DMAP). To extract Φ_{on} , more than 50 individual blinking traces per sample, each from a different QD, are analyzed. Figure S15 shows that the Φ_{on} is significantly larger for the QDs grown with the optimal LB (0.81 vs 0.62). These results suggest that the QDs grown with the optimal LB have less defects than the QDs grown with the suboptimal LB and non-radiative decays are more prominent in the unoptimized sample.

Figure S15. (a,b) Histogram of the blinking ON-time fraction for CdSe/CdS core/shell QDs synthesized from (a) optimal pair ([BBN-SH:4-CF₃Py]) and (b) unoptimal pair ([BBN-SH:DMAP]). (c,d) Representative single QD blinking traces and histogram of the PL intensity for CdSe/CdS core/shell QDs grown with either (c) optimal pair ([BBN-SH:4-CF₃Py]) and (d) unoptimal pair ([BBN-SH:DMAP]). The average ON-time fraction (Φ_{on}) for optimal pair (81%) is higher than one for unoptimal pair (62%).

4. This reviewer does not fully understand why authors stick to spherical QDs. Their new precursors allow temperature independent activations which may be more suited for syntheses of anisotropic NCs. Authors need to compare the shape of NCs by their precursors at different temperatures. For example, shapes of NCs could be compared for the four suboptimal categories in Table 1.

We appreciate this insightful comment. Anisotropic growth, in other words faceted growth, requires thermodynamically controlled growth so that the precursor is preferentially added to the facets that have high surface energy. Thermodynamically controlled growth occurs at a lower temperature range than the

kinetically controlled growth regime. Furthermore, to maintain controlled growth, the reactivity of the precursor should be maintained to be relatively low. In short, to promote anisotropic growth, we will need to use lower growth temperature and the Lewis base that yields low precursor reactivity at the given temperature. Figure S8 shows the QDs grown at the temperature 110°C lower than the melting temperature of the cores, which is much lower than the growth temperature employed for isotropic growth, and using picoline (LB) whose activation temperature is the same as the growth temperature. Even though less spherical QDs were produced, clear evidence of faceted (anisotropic) growth is lacking. We attribute this result to small differences in the surface energy of different facets. To enlarge the difference in surface energy, we added octadecylphosphonic acid, which is known to preferentially bind to the (11 $\bar{2}$ 0) plane of CdS crystals and lowers the surface energy.^{4,5} As expected, strong faceted growth is observed for both wurtzite and zinc-blende CdSe cores (Figure S8). In both cases, the (11 $\bar{2}$ 0) plane is preferably grown, yielding the tetrapod-shaped and rod-like QDs for zinc-blende and wurtzite CdSe cores, respectively.

Figure S8. Both temperature and strongly coordinating ligands are required to achieve the asymmetric QD growth. TEM images (Inset: HRTEM images) of CdSe/CdS QDs synthesized using either wurtzite CdSe QD cores with either (a) [BBN-SH:4-CF₃Py] pair at 200°C, (b) [BBN-SH:picoline] pair at 140°C, or (c) [BBN-SH:picoline] pair at 140°C in the presence of octadecylphosphonic acid. TEM images of CdSe/CdS QDs synthesized using zinc-blende CdSe QD cores with (d) [BBN-SH:picoline] pair at 140°C in the presence of octadecylphosphonic acid. (Scale bar: 20 nm)

(Page 8) Three reaction regimes exist for crystal growth as the reaction temperature varies. At the lowest temperature regime, the added precursor does not have enough energy for surface diffusion, producing particles with irregular shapes. At a higher temperature regime, the added precursor diffuses to the optimal site inducing surface-reaction limited growth that develops thermodynamically preferred facets (Figure S8, see the supporting information (SI) for detail explanation and results). At the highest temperature regime, the surface reaction rate becomes high, yielding kinetically controlled growth and spherical particles.^{44,45} In this paper, we focus on the kinetically controlled growth regime since spherical QDs are preferred in most optoelectronic applications with their superior optical and electronic properties.

Minor comments

1. On page 3, authors claimed “On the other hand, high temperature growth is preferred to promote thermal annealing of defects and kinetically controlled growth, which creates spherical QDs“. Isn't it typically “thermodynamically controlled growth” at high temperature?

Three reaction regimes exist as the reaction temperature varies. At the lowest temperature regime, adatoms do not have enough energy to diffuse to the optimal site, producing particles with irregular shape. At the higher temperature regime, adatoms have enough energy to allow full surface relaxation, inducing thermodynamically controlled growth and producing faceted particles. This regime is also called as surface reaction limited regime. At the highest temperature regime, the difference of the surface energy between different facets becomes less and the rate of surface reaction become high, inducing kinetically controlled growth and mass transport limited growth. In other words, precursors are deposited to all the facets with minimal preference, producing spherical QDs. As expected, spherical QDs are produced at high temperature while lower temperature growth induces either faceted or uncontrolled growth. Currently, spherical QDs are preferred for most optical and optoelectrical applications because it ensures uniform passivation, yielding high QY. Additional explanations and references are added to the main text to further clarify the reaction conditions required for either thermodynamically controlled or kinetically controlled growth.

Figure S8. Both temperature and strongly coordinating ligands are required to achieve the asymmetric QD growth. TEM images (Inset: HRTEM images) of CdSe/CdS QDs synthesized using either wurtzite CdSe QD cores with either (a) [BBN-SH:4-CF₃Py] pair at 200°C, (b) [BBN-SH:picoline] pair at 140°C, or (c) [BBN-SH:picoline] pair at 140°C in the presence of octadecylphosphonic acid. TEM images of CdSe/CdS QDs synthesized using zinc blende CdSe QD cores with (d) [BBN-SH:picoline] pair at 140°C in the presence of octadecylphosphonic acid. (Scale bar: 20 nm)

2. FWHM data of small CdSe cores in Figure 2b are strange. The three sets of experiments start with identical CdSe cores. The FWHMs should be same at 0 minutes for the three sets (b, f, and j in Figure 2).

For all the experiments, we used the identical CdSe cores for both big and small cores. Thus, the FWHM of the initial solutions before heating are the same. The FWHM at 0 minute is different for each experiment because the 0 minute time point refers when the sample reaches the growth temperature. The growth temperature for each precursor is different due to their different reactivity (OT: 240°C, (TMS)₂S: 100°C, BBN-SH with DMAP: 100°C, see Table S4 for detail condition). The melting temperature of small and big CdSe cores are 150°C and 250°C, respectively. Therefore, small cores become unstable at the growth temperature implemented for OT, causing Ostwald ripening and a significant broadening of the emission spectrum as seen in Figure 2b. To prevent the confusion noted by the reviewer, we have modified the Figure 2 to include the time point for before heating. We added the additional note the figure

caption.

Figure 2. [BBN-SH:LB] enables a systematic growth of high quality shells while the QDs produced with conventional precursors show suboptimal quality for all (TMS₂S) or some (OT) of the sizes. (a,e,i) PL spectra of core (dotted line) and core-shell QDs (solid line). (b,f,j) Evolution of FWHM during growth (Δ : the reaction temperature was increased for each shell deposition process, see Table S4 for detail condition), (c,d,g,h,k,l) TEM images (inset: HRTEM images) of (sm)-CdSe/CdS (c,g,k) or (bg)-CdSe/CdS core/shell QDs (d,h,l, scale bar: 20 nm). When using BBN-SH, tight size distribution (FWHM < 25 nm) is obtained for both (sm)- and (bg)-CdSe cores, whereas OT produces QDs with narrow FWHM only from (bg)-CdSe cores. In the case of (TMS)₂S, the emission of the final QDs is broader (FWHM > 32 nm) for both small and big cores. The TEM images confirm narrow size distribution, shape uniformity, and high crystallinity for all size QDs produced with BBN-SH. The structural quality of these QDs is comparable to the QDs grown with OT from big cores.

3. In Figure 2i-j, LBs used should be specified.

As suggested by the reviewer, we added the choice of Lewis base for Figure 2i and j.

<References>

1. Maity P, Debnath T, Ghosh HN. Ultrafast Charge Carrier Delocalization in CdSe/CdS Quasi-Type II and CdS/CdSe Inverted Type I Core–Shell: A Structural Analysis through Carrier-Quenching Study. *J Phys Chem C* 2015, **119**(46): 26202-26211.
2. Dey S, Chen S, Thota S, Shakil MR, Suib SL, Zhao J. Effect of Gradient Alloying on Photoluminescence Blinking of Single CdS_xSe_{1-x} Nanocrystals. *J Phys Chem C* 2016, **120**(37): 20547-20554.
3. Nasilowski M, Spinicelli P, Patriarche G, Dubertret B. Gradient CdSe/CdS Quantum Dots with Room Temperature Biexciton Unity Quantum Yield. *Nano Lett* 2015, **15**: 3953-3958.
4. Manna L, Wang, Cingolani R, Alivisatos AP. First-Principles Modeling of Unpassivated and Surfactant-Passivated Bulk Facets of Wurtzite CdSe: A Model System for Studying the Anisotropic Growth of CdSe Nanocrystals. *J Phys Chem B* 2005, **109**(13): 6183-6192.
5. Rempel JY, Trout BL, Bawendi MG, Jensen KF. Density Functional Theory Study of Ligand Binding on CdSe (0001), (000 $\bar{1}$), and (11 $\bar{2}$ 0) Single Crystal Relaxed and Reconstructed Surfaces: Implications for Nanocrystalline Growth. *J Phys Chem B* 2006, **110**(36): 18007-18016.

<Our response to each reviewer #2's comments (NCOMMS-20-12630-T)>

Reviewer #2:

In the submitted manuscript, Park et al. introduce a new sulfur precursor, whose reactivity can be systematically controlled for the growth of metal chalcogenide QDs. By employing R-B-SH precursors, the authors were able to obtain high-quality QDs at relatively mild reaction temperature. Given the potentially extensive applicability of the precursors, I believe this work is publishable in Nature Communications, if several minor issues are properly addressed.

We appreciate the reviewer for their constructive feedback and comments. We address all of your questions below in details.

1. *It is clear that the optimized precursor and reaction temperature yields high quality CdSe/CdS QDs with high PL QY and narrow emission linewidth. However, the discussions of crystallinity base on inadequate characterizations and speculative arguments. Especially, the slight difference in FWHM of peaks in XRD patterns (Figure S7) does not solely reflect the concentration of defects. The samples in comparison do not have same shape, size, surface configuration, film thickness and roughness.*

We thank the reviewer for pointing this out. We recognize the need for additional characterization data to better probe the crystal quality of QDs. Film thickness and roughness as well as QD size and surface configuration do impact the broadness of XRD peaks. We took necessary precautions to minimize these influences. To prepare similar quality films for XRD measurements, highly experienced researcher with more than 10 years of XRD experience deposited the films using the QD solutions having the same concentration and volume. We also confirm that the QD samples analyzed using XRD have similar sizes and shape. The QDs created with BBN-SH and different LBs have almost identical sizes (9.1 vs 9.2 nm) and shape (circularity: 0.84 vs 0.78). The QDs created with octanethiol also have similar size (9.3 nm) and shape (circularity: 0.82) as other QD samples. In our opinion, the level of difference in shape is not significant enough to cause different levels of peak broadening. To completely eliminate the influence of film thickness, roughness, and morphology, we carried out additional characterization experiments by obtaining the SAED (selected area electron diffraction) patterns using TEM. SAED measurements do not involve film deposition, therefore, are free from the influence of film qualities. Approximately 1000 particles are imaged to obtain the SAED patterns. Due to the random orientation of particles, ring patterns were observed rather than a series of spots observed for single crystalline materials in bulk. After integrating along the radial direction, we compared the FWHM of the (11 $\bar{2}$ 0) peak. We showed that this peak is not affected by the degree of stacking fault *via* simulation. Consistent with our XRD data, the FWHM of this peak is the largest for the mismatched BBN-SH:LB pair. The QDs prepared with octanethiol and the optimized BBN-SH:LB pair show almost identical FWHM, which is significantly smaller than the mismatched pair sample (Figure S13). This result reaffirms our conclusion that upon optimization, BBN-SH grows comparable quality QDs as octanethiol, the precursor known to produce the best quality QDs so far.

Figure S9. Higher precursor reactivity negatively impacts the crystallinity of QDs. (a) XRD data and (b) the summary of the CdS wurtzite(wz) (11 $\bar{2}$ 0) peak (“i”) and the size of coherent volume from (bg)-CdSe/CdS core/shell QDs synthesized by either octanethiol (olive), [BBN-SH:4-CF₃Py] pair (red), or [BBN-SH:DMAP] pair with 12 times slower injection (magenta)

Figure S13. Higher precursor reactivity negatively impacts the crystallinity of QDs. (a-c) Selected area electron diffraction (SAED) pattern (scale bar: 10 (1/nm)), (d) the projected 1D profile along the radius of diffraction ring pattern for resulted QD samples, (e) the FWHM of the arc from the CdS wurtzite(wz) (11 $\bar{2}$ 0) peak (“i”) from (bg)-CdSe/CdS core/shell QDs synthesized by either octanethiol (olive), [BBN-SH:4-CF₃Py] pair (red), or [BBN-SH:DMAP] pair with 12 times slower injection (magenta), and (f-h) the projected 1D profile after baseline correction and peaks with multiple Gaussian fitting.

Figure S14. Stacking fault does not affect on the FWHM of the CdS wurtzite(wz) ($11\bar{2}0$) peak. Simulated XRD data from (a) nanocrystals or (b) bulk comprised with CdS wurtzite crystal (host) and zinc-blende (defect). FWHM of ($11\bar{2}0$) peak (“i”) in simulated XRD data for both nanocrystal (1.4) and bulk (0.1) was maintained while varying the percentage of stacking faults.

(Page 10) To characterize the crystallinity of QDs, we used various characterization techniques (Table S5). Firstly, we analyzed TEM images to examine the fringe pattern of each nanocrystal as described above. Examining TEM images allows visual inspection of the quality of particles but only a small fraction of particles is analyzed. To complement the limitations of TEM analyses, we also analyzed the diffraction pattern of QD samples from X-ray diffraction (XRD) and selected area electron diffraction (SAED). The FWHM of diffraction peaks is positively correlated to the density of defects. To eliminate the influence of stacking faults to the peak broadness, we simulated the XRD profile of CdS nanoparticles having various levels of stacking faults and identified a peak that is not affected by the level of stacking faults (details provided in SI). As shown in Figure S14, the ($11\bar{2}0$) peak is not affected by stacking faults; therefore, it is used for this analysis. Both XRD and SAED results confirm the TEM analysis: the QDs produced with OT and the optimal pair of BBN-SH and LB show similar peak broadness, while the QDs produced with the unoptimized pair of BBN-SH and LB show the significantly broader peak (Figure S9, S13).

2. *Second question on crystallinity discussion is the defining the crystallinity by authors. Authors defined ‘crystallinity’ to ‘the number of particles show clear fringes over the number of particles in TEM images’. How the authors define the clearness of fringes? In addition, the fringes in TEM images are highly affected by the focusing of electron beam to sample in the microscope. Moreover, the extremely low portion of particles is not likely to represent the overall crystallinity of ensemble.*

We recognize the potential concerns regarding using the TEM images to examine the crystal quality of QDs. Even though only limited portion of QD samples can be examined by TEM, TEM images allow us to “visually inspect” the crystal quality of QDs. For TEM, the depth of focus for image plane (S) is > 50 cm for 10K magnification. Therefore, if some particles show clear fringes in the image, the entire particles in that image are in focus. Even in the samples that we see a low % of particles showing clear fringes, every image contained at least a few particles in focus. Here, we explain our strategy to extract representative numbers probing the crystal quality of the ensemble samples in an unbiased manner. Firstly, we inspected > 300 random particles from TEM images. We then count the number of particles that show clear fringes throughout the entire particles. To minimize the ambiguity in the decision, we processed TEM images and increased the contrast. Figure S5 presents how images are processed by step by step and the example particles having clear fringes throughout the particles versus not. To indicate the number of particles examined from TEM images, we added the number particles with clear fringes and the total number of particles counted next to the percentage crystallinity data (ex. 92%, 276/300). The trend of

crystallinity deduced from the TEM analysis is consistent with the newly added SAED pattern data (described details in the previous answer), suggesting the validity of our approach. We also cited a reference paper in main text showing our analysis is a standard procedure.

(Page 7) OT grows high quality core-shell QDs from big cores (diameter(d): 4.6 nm, T_m : 250°C, Figure 2a,b,d). These QDs are highly crystalline (92%, 276/300) and have high QY and narrow FWHM (25 nm).

(Page 7) Furthermore, Figure 2a,e shows that the QDs grown with OT exhibit significantly larger PL shift than those grown with TMS_2S . These results indicate that QD growth at a temperature higher than the T_m of cores leads to particle disintegration⁴¹ and undesired alloying⁴² at the core-shell interface. While TMS_2S produces heterogeneous core-shell structure due to the low temperature growth, the low reactivity of OT requires high temperature growth and yields an alloyed interface. TMS_2S is often used for low temperature growth due to its high reactivity. However, even at very low temperature (100-130°C), its reactivity is still high inducing uncontrolled growth^{29, 32}. As a result, the produced QDs have low crystallinity (45(135/300)-61%(183/300)), broad emission (32-34 nm), non-spherical shape (circularity: <0.5), and low QY (Figure 2e-h). To evaluate the crystallinity of each sample, we counted the percentage of particles showing clear fringes from more than 300 particles that are randomly selected (Figure S5).⁴³

43. Li L, Wang L-L, Johnson DD, Zhang Z, Sanchez SI, Kang JH, et al. Noncrystalline-to-Crystalline Transformations in Pt Nanoparticles. *J Am Chem Soc* 2013, **135**(35): 13062-13072.

Figure S5. Flow of processing TEM images to count the number of QDs crystalline structures. After two steps of processing (a: TEM image without any correction, b: enhancing contrast 10%, c: smoothing), we count the number of QDs with a single crystal (pink circle) out of 300 nanoparticles, while excluding QDs with twin boundary or less clear fringes (green circle).

3. *The discussion on the balance of surface instability and precursor supply is well written and corresponds to the experimental results. However, I consider that the discussion on the facet development during the growth is insufficient and somewhat speculative. I recommend to add more discussion in terms of growth regimes (diffusion or reaction limited).*

We agree with the reviewer that our paper will be stronger with an additional discussion regarding the diffusion (or mass transport) limited versus surface limited reaction regimes. Three reaction regimes exist as the reaction temperature varies. At the lowest temperature regime, adatoms do not have enough energy to diffuse to the optimal site, producing particles with irregular shapes. At the higher temperature regime, adatoms have enough energy to allow full surface relaxation, inducing thermodynamically controlled growth and producing faceted particles. This regime is also called a surface reaction limited regime. At the highest temperature regime, the difference of the surface energy between different facets becomes less and the rate of surface reaction becomes high, inducing kinetically controlled growth and mass transport limited growth.^{1, 2} In other words, precursors are deposited to all the facets with minimal preference, producing spherical QDs. As expected, spherical QDs are produced at high temperatures while lower

temperature growth induces either faceted or uncontrolled growth. Currently, spherical QDs are preferred for most optical and optoelectrical applications because it ensures uniform passivation, yielding high QY. Additional explanations and references are added to the main text to further clarify the reaction conditions required for thermodynamically controlled vs kinetically controlled growth. Additional explanations and references are added to the main text to further clarify the reaction conditions required for thermodynamically controlled vs kinetically controlled growth.

We also added new data showing the facet development at low temperature growth. Anisotropic growth, in other words faceted growth, requires thermodynamically controlled growth so that the precursor is preferentially deposited to the facets that have high surface energy. Furthermore, to maintain controlled growth, the reactivity of the precursor should be maintained to be relatively low. Therefore, we used lower growth temperature than kinetically controlled growth and the Lewis base that yields low precursor reactivity at the given temperature. Figure S8 shows the QDs grown at the temperature 110°C lower than the melting temperature of the cores, which is much lower than the growth temperature employed for isotropic growth, and using 4-CF₃Py (LB) whose activation temperature is the same as the initial growth temperature. Even though less spherical QDs were produced, clear evidence of faceted (anisotropic) growth is lacking. We attribute this result to small differences in the surface energy of different facets. To enlarge the difference in surface energy, we added octadecylphosphonic acid, which is known to preferentially bind to the (11 $\bar{2}$ 0) plane of CdSe crystals and lowers the surface energy.^{3, 4} As expected, strong faceted growth is observed for both wurtzite and zinc-blende CdSe cores (Figure S8). In both cases, the (11 $\bar{2}$ 0) plane is preferably grown, yielding the tetrapod-shaped and rod-like QDs for zinc-blende and wurtzite CdSe cores, respectively.

(Page 8) Three reaction regimes exist for crystal growth as the reaction temperature varies. At the lowest temperature regime, the added precursor does not have enough energy for surface diffusion, producing particles with irregular shapes. At a higher temperature regime, the added precursor diffuses to the optimal site inducing surface-reaction limited growth that develops thermodynamically preferred facets (Figure S8, see the supporting information (SI) for detail explanation and results). At the highest temperature regime, the surface reaction rate becomes high, yielding kinetically controlled growth and spherical particles.^{44,45} In this paper, we focus on the kinetically controlled growth regime since spherical QDs are preferred in most optoelectronic applications with their superior optical and electronic properties.

Figure S8. Both temperature and strongly coordinating ligands are required to achieve the asymmetric QD growth. TEM images (Inset: HRTEM images) of CdSe/CdS QDs synthesized using either wurtzite CdSe QD cores with either (a) [BBN-SH:4-CF₃Py] pair at 200°C, (b) [BBN-SH:picoline] pair at 140°C, or (c) [BBN-SH:picoline] pair at 140°C in the presence of octadecylphosphonic acid. TEM images of CdSe/CdS QDs synthesized using zinc-blende CdSe QD cores with (d) [BBN-SH:picoline] pair at 140°C in the presence of octadecylphosphonic acid. (Scale bar: 20 nm)

4. *Though the authors introduced the synthesis of CuInS₂ NCs using R-B-SH precursors, the main data consist of Cd and Pb based materials. I recommend to emphasize the extensibility of the authors' approach to other semiconductor NCs, such as metal selenides or pnictides.*

We are currently working on extending the chemistry to other type of nanoparticles such as metal selenides and pnictides as the reviewer mentioned. We have added a few sentences discussing the extensibility of our new precursor design to other materials.

(Page 15) Our result emphasizes the benefit of employing a chemically modulatable bond in nanoparticle precursors. The QD growth principle established here inspires the design of new anion precursors for other types of nanoparticles, including metal chalcogenides and pnictogenides QDs. This study provides a foundation for the controlled growth of high quality nanocrystals.

<References>

1. Kappes BB, Leong GJ, Gilmer GH, Richards RM, Ciobanu CV. Metallic nanocrystals synthesized in solution: a brief review of crystal shape theory and crystallographic characterization. *Cryst Res Technol* 2015, **50**(9-10): 801-816.
2. Ohring M. Chapter 6 - Chemical Vapor Deposition. In: Ohring M (ed). *Materials Science of Thin Films (Second Edition)*. Academic Press: San Diego, 2002, pp 277-355.
3. Manna L, Wang, Cingolani R, Alivisatos AP. First-Principles Modeling of Unpassivated and Surfactant-Passivated Bulk Facets of Wurtzite CdSe: A Model System for Studying the Anisotropic Growth of CdSe Nanocrystals. *J Phys Chem B* 2005, **109**(13): 6183-6192.
4. Rempel JY, Trout BL, Bawendi MG, Jensen KF. Density Functional Theory Study of Ligand Binding on CdSe (0001), (000 $\bar{1}$), and (1120) Single Crystal Relaxed and Reconstructed Surfaces: Implications for Nanocrystalline Growth. *J Phys Chem B* 2006, **110**(36): 18007-18016.

REVIEWERS' COMMENTS

Reviewer #1 (Remarks to the Author):

This manuscript has been revised with great care and attention to the reviewers' comments. It merits publication without further revision. However, there is one issue that needs to be resolved. Authors use the terms 'thermodynamically controlled growth' and 'kinetically controlled growth' quite contrary to conventional ways. Conventionally 'thermodynamically controlled growth' is a growth that yields more thermodynamically preferred product which would be spheres in colloid syntheses. Authors are consistently using the terms oppositely. For an example, authors stated "At the highest temperature regime, the surface reaction rate becomes high, yielding kinetically controlled growth and spherical particles". Authors need to properly define the terms if they wish to use them other than conventional ways.

Reviewer #2 (Remarks to the Author):

The revised manuscript by the group of Dr. Han appears to address all of the reviewer comments raised in the previous round. The manuscript now suits the readership of Nature Communications. Probably one minor comment on the authors' response: in the response letter, the authors claim that, if I may paraphrase, no systematic work has been reported to grow high-quality crystals. I think this statement, although not included in the narrative of the manuscript, is an overkill. The authors did not need to disparage the collective efforts of the academic society to show off the results. There have been numerous attempts to grow highly crystalline QDs. In fact, I am not sure how the authors defined "systematic", because compared to previous studies referenced in the manuscript, I do not find the approaches convincingly more systematic.

On a probably related note, I do not question the experience of the researcher in charge of sample preparation. But, the width of the XRD peaks is not a smoking gun. In fact, the patterns in Figure S9 are rather unconvincing.

Beside these minor points, the manuscript seems good to go. Great job, Dr. Han and fellows!

<Our response of Reviewers' comments>

Reviewer #1 (Remarks to the Author):

This manuscript has been revised with great care and attention to the reviewers' comments. It merits publication without further revision. However, there is one issue that needs to be resolved. Authors use the terms 'thermodynamically controlled growth' and 'kinetically controlled growth' quite contrary to conventional ways. Conventionally 'thermodynamically controlled growth' is a growth that yields more thermodynamically preferred product which would be spheres in colloid syntheses. Authors are consistently using the terms oppositely. For an example, authors stated "At the highest temperature regime, the surface reaction rate becomes high, yielding kinetically controlled growth and spherical particles". Authors need to properly define the terms if they wish to use them other than conventional ways.

We have added more explanation of 'thermodynamically controlled growth' and 'kinetically controlled growth' and addressed the issue that the reviewer has raised. Now, our usage of terms are consistent with the conventional usage.

(page 3) On the other hand, high temperature growth is preferred to promote thermal annealing of defects and create spherical QDs.^{5,25}

(page 10) As the reaction temperature approaches the melting temperature of nanoparticles, the energy difference between facets decreases, creating spherical particles even with thermodynamically controlled growth.^{3, 46, 47} In this paper, we focus on the third regime since spherical QDs are preferred in most optoelectronic applications with their superior optical and electronic properties.

(page 14) Close examination of the growth results provides insight into a nanoparticle growth process.

Reviewer #2 (Remarks to the Author):

The revised manuscript by the group of Dr. Han appears to address all of the reviewer comments raised in the previous round. The manuscript now suits the readership of Nature Communications. Probably one minor comment on the authors' response: in the response letter, the authors claim that, if I may paraphrase, no systematic work has been reported to grow high-quality crystals. I think this statement, although not included in the narrative of the manuscript, is an overkill. The authors did not need to disparage the collective efforts of the academic society to show off the results. There have been numerous attempts to grow highly crystalline QDs. In fact, I am not sure how the authors defined "systematic", because compared to previous studies referenced in the manuscript, I do not find the approaches convincingly more systematic.

On a probably related note, I do not question the experience of the researcher in charge of sample preparation. But, the width of the XRD peaks is not a smoking gun. In fact, the patterns in Figure S9 are rather unconvincing.

Beside these minor points, the manuscript seems good to go. Great job, Dr. Han and fellows!

Regarding the first point, we absolutely agree with the reviewer. We toned down our explanation on the limitations of the previous approaches by 1) removing the sentences saying “no systematic method exist to growth high quality QDs” and 2) updating the previous sentences as follows.

(page 1) Although the synthetic strategies for size control are well established and widely applicable to various QD systems, the strategies to tune the structural characteristics of QDs, such as morphology, and crystallinity, are mostly material specific.

(page 3) Despite the importance of the structural quality, no universal method yet exists to grow highly crystalline, spherical QDs having a desired core-shell interface structure.

(page 7) Secondly, no systematic method exists to identify a precursor that yields an optimal surface reaction rate at a specific reaction temperature, leading to a trial-and-error based selection of precursors.

Regarding the second comments on XRD: Since the size of QDs is too small, the XRD patterns did not show a dramatic difference between QD samples. However, the results from different characterization methods (both structural properties (XRD, SAED, TEM) and optical properties (QY, single QD blinking measurement)) support our claim that CdSe/CdS core/shell QDs from the optimized pair of BBN-SH and LB showed a higher quality shell growth than those from the mismatched pair. Therefore, even though XRD measurements alone cannot definitively prove our points, all our experimental results collectively support our claims. Additionally, we used a computer-aided automatic Gaussian-fitting and background correction to analyze both XRD and SAED patterns to eliminate human interpretentaion of the data.